# Simultaneous Detection of *NF1*, *SPRED1*, *LZTR1*, and *NF2* Gene Mutations by Targeted NGS in an Italian Cohort of Suspected NF1 Patients

**DOI:** 10.3390/genes11060671

**Published:** 2020-06-19

**Authors:** Donatella Bianchessi, Maria Cristina Ibba, Veronica Saletti, Stefania Blasa, Tiziana Langella, Rosina Paterra, Giulia Anna Cagnoli, Giulia Melloni, Giulietta Scuvera, Federica Natacci, Claudia Cesaretti, Gaetano Finocchiaro, Marica Eoli

**Affiliations:** 1Molecular Neuro-Oncology Unit, Fondazione IRCCS Istituto Neurologico Carlo Besta, via Celoria 11, 20133 Milan, Italy; donata.bianchessi@istituto-besta.it (D.B.); mariacristina.ibba@istituto-besta.it (M.C.I.); s.blasa@campus.unimib.it (S.B.); tiziana.langella89@gmail.com (T.L.); rosina.paterra@istituto-besta.it (R.P.); gaetano.finocchiaro@istituto-besta.it (G.F.); 2Developmental Neurology Unit, Fondazione IRCCS Istituto Neurologico Carlo Besta, via Celoria 11, 20133 Milan, Italy; veronica.saletti@istituto-besta.it (V.S.); giulia.melloni@istituto-besta.it (G.M.); 3Department of Biotechnology and Biosciences, University of Milano-Bicocca, Piazza dell’Ateneo Nuovo, 1, 20126 Milan, Italy; 4Molecular Immunology Unit, Department of Research, Fondazione IRCCS Istituto Nazionale dei Tumori, Via Venezian, 20133 Milan, Italy; 5Medical Genetics Unit, Woman-Child-Newborn Department, Fondazione IRCCS Ca’ Granda-Ospedale Maggiore Policlinico, via Francesco Sforza 28, 20122 Milan, Italy; giulia.cagnoli@policlinico.mi.it (G.A.C.); federica.natacci@policlinico.mi.it (F.N.); claudia.cesaretti@policlinico.mi.it (C.C.); 6Pediatric Highly Intensive Care Unit, Università degli Studi di Milano, Fondazione IRCCS Ca’ Granda Ospedale Maggiore Policlinico, via Francesco Sforza 28, 20122 Milan, Italy; giulietta.scuvera@policlinico.mi.it

**Keywords:** neurofibromatosis type 1, targeted next generation sequencing (NGS), *NF1*, *SPRED1*, *LZTR1*, *NF2* genes pathogenic variants, schwannomatosis

## Abstract

Neurofibromatosis type 1 (NF1) displays overlapping phenotypes with other neurocutaneous diseases such as Legius Syndrome. Here, we present results obtained using a next generation sequencing (NGS) panel including *NF1*, *NF2*, *SPRED1*, *SMARCB1*, and *LZTR1* genes on Ion Torrent. Together with NGS, the Multiplex Ligation-Dependent Probe Amplification Analysis (MLPA) method was performed to rule out large deletions/duplications in *NF1* gene; we validated the MLPA/NGS approach using Sanger sequencing on DNA or RNA of both positive and negative samples. In our cohort, a pathogenic variant was found in 175 patients; the pathogenic variant was observed in *NF1* gene in 168 cases. A *SPRED1* pathogenic variant was also found in one child and in a one year old boy, both *NF2* and *LZTR1* pathogenic variants were observed; in addition, we identified five *LZTR1* pathogenic variants in three children and two adults. Six *NF1* pathogenic variants, that the NGS analysis failed to identify, were detected on RNA by Sanger. NGS allows the identification of novel mutations in five genes in the same sequencing run, permitting unambiguous recognition of disorders with overlapping phenotypes with NF1 and facilitating genetic counseling and a personalized follow-up.

## 1. Introduction

Neurofibromatosis type 1 (NF1), an autosomal dominant disorder, is the most frequent tumor predisposition syndrome [1].

The condition is caused by mutations in the neurofibromin gene (17q11.2.5-7) encoding a negative regulator of Ras guanosine triphosphate (GTPase) proteins and acting as a tumor suppressor gene. Mutation detection in the *NF1* gene is challenging, due to the large size of the gene (>350 kb), the presence of pseudogenes, the lack of hot spots, and the high mutation rate that is responsible for 50% of all pathogenic variants being de novo [1,2].

Point mutations, including nonsense, missense, frameshift, and splicing mutations, are the most frequent alterations identified in *NF1* (around 80–90%) [3,4]. Only 5% of NF1 patients have deletions of the entire *NF1* gene including contiguous genes (1–1.4 Mb) [4,5]. Insertion, duplication, and copy number variations (CNV) are rarely reported in the literature [6]. In the Leiden Open Variation Database-NF1, over two thousand unique germline *NF1* variants [7] are reported. Several studies using a comprehensive approach that includes RNA analysis coupled with Multiplex Ligation-Dependent Probe Amplification Analysis (MLPA) reported a germline mutation rate in 83−95% of clinically confirmed cases [8,9,10].

The diagnosis of NF1 was based on the clinical criteria defined in a National Institutes of Health (NIH) conference in 1988 [11] requiring the presence of at least two specific features of the disease. In most cases, a definitive clinical diagnosis can be performed. However, disease signs are age-dependent and the full clinical manifestation usually appears at 8 years of age. Furthermore, in 2007 a clinically overlapping disorder, Legius Syndrome, characterized by the presence of multiple café au lait spots (CALS), freckling and macrocephalia, was described [12]. In a large database of individuals that met NIH criteria for NF1 diagnosis 1.9% had a molecular diagnosis of Legius [13] and 8% of cases aged 0–20 years with CAL but without non-pigmentary criteria for NF1 had *SPRED1* mutations [8].

Spinal Neurofibromatosis (SNF), a distinct clinical entity of NF1, is characterized by bilateral neurofibromas involving all spinal roots and a few, if any, cutaneous manifestations [14]. Most of those patients received a delayed NF1 diagnosis, because the NIH diagnostic criteria were unfulfilled. In addition, a *SOS1* mutation was identified [15] in two familial cases with previous clinical diagnosis of NF1 and multiple spinal nerve enlargements resembling plexiform neurofibromas. Individuals with constitutional mismatch repair (MMR) deficiency, a rare tumor predisposing syndrome caused by biallelic mutations in one of *MMR* genes, display features reminiscent of NF1 [16].

Therefore, the recognition of disorders with overlapping clinical and radiological phenotypes but with different prognoses, emphasize the importance of molecular diagnosis.

Targeted Next Generation Sequencing (NGS) is now applied to the fast and unambiguous diagnosis of NF1, schwannomatosis or Legius Syndrome. In the present study, we validated an NGS approach coupled with MLPA, analyzing prospectively 250 consecutive patients with suspected NF1.

## 2. Materials and Methods

### 2.1. Patient Population

Two-hundred fifty consecutive patients, referred to our institution from several Italian centers for suspected NF1 from 1 July 2017 to 30 June 2018, were included in the study. The median age was 15.5 years (3 months–74 years), 132 were pediatric (<18 years) and 118 adult cases; females and males were equally represented. One-hundred eighty-four were sporadic patients and 66 familial cases. One-hundred seventy-one fulfilled NIH criteria (68%), 82/184 sporadic subjects had just pigmentary criteria (i.e., CAL with or without freckling and no other NF1 features,) and age < 20 years. Ninety-one percent of patients manifesting only one clinical feature of NF1 (79 cases) were children, 65% were under 8 years. All patients or authorized relatives gave informed consent prior to genetic analysis. The investigations were carried out in accordance with the principles laid down in the 2013 revision of the Declaration of Helsinki. This retrospective study was approved by the Fondazione IRCCS Istituto Neurologico Carlo Besta Ethical Committee and Scientific Board (N°72-2020).

### 2.2. DNA and RNA Extraction and Retro-Transcription

As already described by Bianchessi [17], genomic DNA (gDNA) was isolated from blood samples in ethylenediamine tetraacetic acid (EDTA) using a Gentra Puregene^®^ Blood Core Kit B (Qiagen, Venlo, The Netherlands; Carlsband, CA, USA).

RNA samples were collected in Tempus^TM^ Blood RNA Tubes (Life Technologies, Carlsbad, CA, USA) and extracted with a Tempus^TM^ Spin RNA Isolation Kit within 5 days. DNase treatment with Absolute RNA Wash Solution was performed for all samples during the RNA extraction protocol. RNA samples were reverse-transcribed using 50 units of High-Capacity cDNA Reverse Transcription mix (Life Technologies) and 20 units of RNAse inhibitor (Ambion, Austin, TX, USA). β-2-Microglobulin amplification was used as a quality control for retro-transcription.

### 2.3. Multiplex Ligation-Dependent Probe Amplification Analysis (MLPA)

Patients’ DNA was analyzed by MLPA with NF1 MLPA salsa P081 and P082 (MRC Holland, Amsterdam, The Netherlands). P081/P082 salsa kit identified single- and multi-exon deletion/duplications inside the *NF1* gene. The amplification products were covering all 58 exons of the *NF1* gene. P081/P082 positive patients for the entire *NF1* deletion were screened also with MLPA P122 salsa kit. Results obtained by ABI Prism 3130 Genetic Analyzer (Life Technologies) were analyzed with the Coffalyser.Net Software (MRC Holland, Amsterdam, The Netherlands).

Patients positive for deletion or duplication were excluded from NGS analysis.

### 2.4. NGS Sequencing

Library preparation was carried out using the Ion AmpliSeq Library Kit 2.0 (Life Technologies). Thirty ng of gDNA were added to multiplex primer pools to amplify target genomic regions. Primers were partially digested using a FuPa reagent, and the sequencing adapters were ligated to the amplicons. The library was purified using the Agencourt AMPure XP reagent (Beckmann Coulter, CA, USA). Concentrations and quality of purified libraries, as well as size of the amplicon, were determined using Qubit^®^ 2.0 (Life Technologies) and Agilent 2100 Bioanalyzer^®^ (Agilent Technologies, Inc., Santa Clara, CA, USA) with Agilent^®^ High Sensitivity DNA kit (Agilent Technologies, Inc., Santa Clara, CA, USA). Template preparation was performed with Ion PGM™ Hi-Q™ OT2 Kit (Cat. no. A27739) (Thermo Fisher Scientific, Waltham, MA, CA, USA) on Ion One Touch instrument using the emulsion PCR method.

Unlinked beads were removed from the solution during the semi-automated enrichment process on Ion One Touch ES instrument (Life Technologies).

Libraries were pooled equimolar and after adding the sequencing primer and DNA polymerase, the fully prepared Ion Sphere Particle (ISP) beads were loaded into an Ion 318 sequencing chip and the sequencing runs were performed using the Ion PGM™ Hi-Q™ Sequencing Kit (Life Technologies) with 500 flows.

The multigenic panel was designed using Ampliseq Designer (Life Technologies) and included all exons and 3′/5′-UTR of the *NF1*, *LZTR1*, *NF2*, *SMARCB1*, and *SPRED1* genes.

The average coverage was 96% and the target regions were sequenced at a 130× depth.

### 2.5. Data Analysis 

Data of runs were processed using the Ion Torrent Suite 5.0 VariantCaller 5.0, Coverage Analysis 5.0 (Life Technologies) and the Ion Reporter (Life Technologies).

The TMAP algorithm was used to align the reads to the hg19 human reference genome, and the variant caller plug-in was selected to run the search for germ line variants in the targeted regions.

To visualize the status of each read alignment and variant interpretation the Integrative Genomic Viewer version 2.3 (IGV) (Broad Institute and the Regents of the University of California) was used.

We removed all the common variants (Minor Allele Frequency, MAF > 1%) reported in the following public databases: *1000 Genomes Project*. 

The effect on genes and proteins of the mutations identified were predicted based on Mutation Taster HGMD (Human Genome Mutation Database–Institute of Medical Genetics, Cardiff, Wales, UK; and LOVD [18] databases were interrogated to verify if the mutation was novel.

Novel variants with amino acid changes were further examined for their disease-causing potential using PolyPhen-2 [19]. The possible effects on mRNA (canonical and not canonical splicing mutation) were evaluated with Splice site Prediction by neural network [20], the Human Splicing Finder [21] and ESE Finder tools [22,23].

### 2.6. Sanger Sequencing

Called and deleterious variants identified with NGS Ion Torrent were confirmed by Sanger Sequencing. PCR reactions were performed using Taq Gold Polymerase^®^ (Life Technologies, Foster City, CA, USA). The size of PCR products was verified by electrophoresis on a 2% agarose gel. The PCR products were purified using ExoSAP-IT^®^ (USB Corporation, Cleveland, OH, USA) according to the manufacturer’s protocol and were sequenced in both directions using the BigDye terminator sequencing kit v1.1 (Life Technologies) on ABI 3130 Genetic Analyzer (Life Technologies). 

Sequencing analysis on Sanger using the BigDye terminator sequencing kit v3.1 (Life Technologies) was performed on negative *NF1* NGS samples using cDNA that was amplified in 23 overlapping fragments from 400 to 560 bp. 

### 2.7. Submission of Genomic Variations

All novel pathogenic variants identified in *NF1*, *NF2*, *SPRED1* and *LZTR1* genes have been deposited in the “Leiden open (source) variation database” (LOVD) public database [18]. 

## 3. Results

A comprehensive analysis of *NF1*, *NF2*, *SPRED1*, *SMARCB1*, and *LZTR1* genes was performed on a total of 250 consecutive cases with suspected NF1 using the NGS panel. Before NGS, MLPA method was performed to rule out large deletions/duplications in the *NF1* gene. Screening results are summarized in Figure 1.

Our approach allowed the identification of a pathogenic variant in 175 patients: 168 in *NF1*, 1 in *SPRED1*, 6 in *LZTR1* and 1 in *NF2* genes, two variants were present in the same patient and six benign variations (5 in *NF1* and 1 in *LZTR1* genes).

### 3.1. Detection of NF1 Pathogenic Variants

Using NGS coupled with MLPA and cDNA Sanger sequencing, we identified pathogenic *NF1* variants in 168/250 unrelated patients submitted for NF1 clinical genetic testing to the Neurological Institute C. Besta (Figure 1). 138 mutations were detected among the 171 fulfilling NIH criteria for NF1 diagnosis (detection rate 81%), while 30 pathogenic *NF1* variants were detected (21 by NGS and 7 by MLPA) in 79 cases not fulfilling NIH clinical criteria (37.9%). Furthermore, 39 of 82 sporadic subjects with CAL and only pigmentary criteria for NF1 (age < 20 years) had *NF1* pathogenic variants (47%), 3 had *LZTR1* pathogenic variants (3.6%); in one of them a *NF2* pathogenic variant (1.1%) was also present.

Fifty-two of 160 pathogenic variants (31%) (40 detected by NGS and 12 by MLPA) were novel, i.e., not present in HGMD and LOVD databases. Novel pathogenic variants were deposited in the LOVD database [18] and described according to recommendations of the Human Genome Structural Variant (HGSV) consortium.

The 52 novel pathogenic variants are reported in Table 1.

Four subtypes of gross *NF1* gene deletions have been described that differ in terms of the deletion size and the positions of their respective breakpoints: type 1, type 2, type 3, and atypical *NF1* deletions [3]. Type-1 microdeletion is characterized by breakpoints located within the low copy number repeats *NF1-REPa* and *NF1-REPc* and involves the *NF1* gene and contiguous genes *SUZ12P*, *CRLF3*, *ATAD5, ADAP2*, *RNF135*, *UTP6*, *SUZ12* and *LRRC37B*; while atypical *NF1* microdeletions have non-recurring breakpoints [13].

MLPA analysis of the *NF1* gene identified 28/168 pathogenic variations: four type 1 microdeletions, two atypical microdeletions, seven intragenic deletions and two intragenic duplication. In addition MLPA revealed also six splicing pathogenic variants, two nonsense and five frameshift pathogenic variants that were located in the probe binding sequence.

### 3.2. Characterization of the Pathogenic Variants Detected by NGS

We observed the following types of pathogenic variants: frameshift (31% 41/132), in frame deletions (2.2% 3/132), splicing (27% 36/132), missense (16% 22/132) and nonsense (23% 30/132) pathogenic variants). Overall, 115 (87%) pathogenic variants were “private” (only observed in one unrelated patient/family). The remaining pathogenic variants were found more than once.

NGS analysis also allowed to detect five benign variations of the *NF1* gene in three subjects. The first one was missense variation c.4768 C>T (*p*. Arg1590Trp) also observed in the father and the brother who did not show signs of the disease. Another missense benign variant was c.3734 C>G (*p*.Thr1245Ser) inherited from the father and the paternal grandmother who did not show clinical phenotype. In the same subject the variant c.861 C>T (*p*.Asp287Asp) (described as polymorphism; the variation c.2326-25 C>G and the variation c.7394 + 61 A>C, all resulted by in silico prediction analysis with “Mutation Taster” as benign.

All *NF1* variants identified by NGS were confirmed by Sanger DNA sequencing. 

All the *NF1* NGS negative cases (*n*° = 90) were tested using cDNA Sanger sequencing. The NGS analysis failed to identify six types of pathogenic variant in eight patients that were detected on RNA by Sanger: these pathogenic variants are reported on Table 2. Four were frameshift and two were splicing.

By NGS, a *SPRED1* pathogenic variant that was previously described, [12] (c.349 C>T causing a premature stop codon), was observed in a five year old child, with macrocephaly, diffuse Cal spots, freckling, no neurofibroma, no Lisch nodule neither coroideal amartoma; the father had the same clinical phenotype. Legius Syndrome was diagnosed in both.

Furthermore, a *NF2* missense pathogenic variant already reported [24] c.397 T>C, p.Cys133Arg and a de novo *LZTR1* nonsense alteration c.844 C>T, *p*.Gln282* were both found in a one year old child with only eight Cal spots.

The de novo frameshift *LZTR1* pathogenic variant: c.154_154 delC, p.His52lle Fs*49 was observed in a 12 years old girl with only 15 Cal spots and her father with multiple Schwannomas; the de novo missense pathogenic variant c.1394 C>T, Ala465Val was detected in a six years old boy with diffuse freckling, pectum excavatum, scleral nevi. Further genetic analysis showed the heteroziygous pathogenic variant c.1403 C>T in PTPN11 gene allowing Leopard Syndrome to be diagnosed. The already described missense pathogenic variant: c.1394 C>A, Ala465Glu missense-in a 6 years girl with only ten Cal spots (see Table 1).

In a 55 years old female, the de novo nonsense pathogenic variant *LZTR1* c.161 G>A was detected, the patient showed only two Cal spots, multiple subcutaneous nodules in limbs and trunk, and neither Lisch nodule nor coroideal amartomas. At cervical MRI an intramedullary alteration in T2 weighted imagines was present.

The de novo intronic splicing *LTZTR1* pathogenic variant: c.264-3 A>G, r.264_320del57, p.Lys89Arg107del was detected in a 25 years old male with two Cal spots, subcutaneous nodules, retroperitoneal nodules, D12-L1 nodules, neither Lisch nodule nor coroideal amartoma. After genetic testing, the diagnosis of schwannomatosis was confirmed according to current diagnostic criteria, when one subcutaneous nodule and the D12 nodule were examined respectively. The histopathological diagnosis was Schwannoma for both.

A synonymous single base *LZTR1* variation c.1530 C>T r.1530 C>U, p.His510His likely not pathogenic was observed in a 55 year old man, one Cal spot, one angiolipoma, multiple paraspinal nodules at lumbar spine involving also left iliopsoas, and neither Lisch nodule nor coroideal amartoma.

No variations were observed in the *SMARCB1* gene.

## 4. Discussion

*NF1* pathogenic variants can be detected through several different techniques. 

To date, the application of NGS method for the molecular diagnosis of NF1 has been reported in six studies. Different NGS platforms have been used, some studies were focused only on the *NF1* gene [25,26], while others included also genes such as *SPRED1* [27,28] *BRAF*, *p53* [29]. Apart from Pasmant [27], in all studies targeted NGS was combined with MLPA to rule out large deletions/duplications in *NF1* gene.

Using NGS coupled with MLPA, our detection rate, for *NF1* pathogenic variants (77%) is similar to that reported (76%) in Turkish patients [30] and lower than that found in other reports ranging from 88% to 96% [25,26,27]. The discrepancies can reflect disparities in the inclusion criteria of the patients studied. We and [30] tested all consecutive patients referred for suspected neurofibromatosis type 1 to our laboratories over a certain interval time without any selection of the cases.

NGS is a complex technology, requiring validation efforts. Various variables related to laboratory procedures and bioinformatics analysis can influence the accuracy of the results.

For clinicians it is important to test accuracy and to understand the potential limitations of the sequencing technologies employed. 

Unfortunately all the negative cases were tested using DNA and/or cDNA Sanger sequencing only in another study [27].

No *NF1* unidentified pathogenic variant was found in the French study [26], while we observed eight cases in which both MLPA and NGS failed to identify the pathogenic variants. In two cases these pathogenic variants were intronic (c.288 + 1138 C>T; c.7907 + 791 C>G) therefore not covered in our panel, two pathogenic variants were inserted just before repeated C or A sequences (c.2033 dupC, c.4224_4225 del AA insT) that can interfere with probe ligation. Furthermore, it is already known that NGS approaches may present limitations in detection of insertion/deletion. The most frequently undetected pathogenic variant was in exon five: a tetranucleotide tandem repeat (TGTTTGTT) comprising nucleotides 495–502 that can prevent efficient ligation of MLPA and NGS probes is present in this exon [31].

In addition, Calì [26] reported that NGS analysis failed to identify a pathogenic variant on the same exon that was detected on DNA by Sanger.

However, using MLPA followed by NGS and cDNA amplification and sequencing we have been able to find the disease-causing lesion in 138/171 (81%) of our patients fulfilling NIH criteria for NF1 diagnosis. Based on our results, we suggest performing cDNA analysis by Sanger sequencing in all patients satisfying diagnostic criteria with negative NGS/MLPA testing for NF1.

Among the 28 patients fulfilling NF1 NH criteria with no *NF1*, *SPRED1* and *LZTR1* pathogenic variant, five patients were affected by a sporadic segmental NF1. The typical features of NF1 were restricted to one part of the body and both parents were unaffected [32]. This distinct form of NF1 has an estimated prevalence equal to 0.002% in the general population [32,33], 10–20 times lower than the frequency of generalized NF1. In those case *NF1* pathogenic variant can be present in a very low percentage of blood cells [32,34,35] and therefore not detectable by NGS at a 130X depth.

Moreover, three negative patients had a spinal form of NF1 and recently Santoro [15] has reported p.Ser548Arg missense pathogenic variant in *SOS1* gene in a patient with bilateral cervical and lumbar spinal lesions resembling plexiform neurofibromas and features suggestive of NF1. We therefore cannot exclude that these three patients bear a causative pathogenic variant of *SOS1* or in other genes not tested with this panel.

The possible occurrence of pathogenic variants in “deep” intronic regions, reported in NF1 cases [36], could be at the basis of the clinical NF1 diagnosis in 14 of the negative patients, who showed a mild sporadic form of NF1 including Cal and freckling, and six others who showed a typical NF1 familial form. 

Because of the overlapping phenotype of Legius Syndrome with *NF1*, the *SPRED1* gene was included in our panel. The exact prevalence of this syndrome is still unknown, using NGS we observed one *SPRED1* pathogenic variant in one case out of 171 (0.58%) patients fulfilling diagnostic criteria for NF1. In other studies this percentage ranged from 1.9% [13] to 4% [27]. Recently the co-occurrence of pathogenic variants in the *NF1* and *SPRED1* genes was observed using NGS in one family with NF1 and Legius Syndrome [37]. One sibling with typical features of NF1 with microdeletion type 1 inherited a complete deletion of the *NF1* gene from her mother and carried a variant of unknown significance in the *SPRED1* gene ac.944C>T, p.(Pro315Leu).

Previously, one more case of co-occurrence of NF1 and Legius Syndrome in the same family (with confirmed pathogenic variants in the *NF1* and *SPRED1* genes) was reported [13]. Those findings highlight the relevance of the simultaneous detection of *NF1* and *SPRED1* in genetic testing and the complexity of molecular diagnosis in patients suspected NF1.

Schwannomatosis (SCHW) is a rare disease predisposing to multiple peripheral neurologic tumors development with an incidence approximately of one in 69,000, even if the true incidence could be higher due to the difficulties in case ascertainment. To date two causative genes for schwannomatosis, *SMACRB1* and *LZTR1*, were identified [38,39]. Approximately, one third of patients with Schwannomatosis are carriers of a germline pathogenic variant in *LZTR1* (Leucin Zipper Transcription Regulator 1).

The clinical features of the disease are still poorly described, especially when *LZTR1* is mutated. Most patients developed symptoms in the second or third decade of life but the diagnosis is usually delayed for several years. The most common presenting features is asymtomatic mass or diffuse or localized pain. Until recently, the presence of vestibular schwannoma excluded a diagnosis of schwannomatosis but patients with *SMARCB1* and *LZTR1* pathogenic variants that met neurofibromatosis type 2 (NF2) criteria have been described [40,41]. However, recently, pathogenic variants in *LZTR1* have also been involved in a small proportion of patients with Noonan syndrome, a rare neurodevelopmental syndrome [42] either in a dominant or in a recessive fashion of transmission; it has been postulated that dominant negative missense variant cause the dominant form and hypomorphic in trans with loss-of-function variants are at the base of the recessive one. Moreover, a recurrent mixed glioma tumor of oligoastrocytoma type was described in such a patient [43] and coexistence of schwannomatosis and glioblastoma in two families [44].

Very recently the role of LZTR1 as a negative RAS modulator has been demonstrated, since different classes of *LZTR1* mutations are predicted to interfere with ubiquitination and degradation of substrates that act as a positive modulator of the RAS-MAPK pathway [45]. 

Even if the presence of cutaneous Cal spots is included among clinical signs in Rasopathies, until now they are not considered a typical sign of schwannomatosis due to LZTR1 pathogenic variants.

In our patients, the presence of Cal spots and the young age have led to the suspicion of NF1. The exams which followed confirmed the diagnosis of SCHW in the two adult patients according to actual diagnostic criteria. The other patients were too young at present to confirm NF2 and SCHW diagnosis.

However, the observation of more than 6 Cals in young patients bearing a pathogenic variant in *LZTR1* or *NF2* genes is interesting, and deserves to be explored in further studies.

Because of the clinical overlap of NF1 and other phenotypes, the variability of the disease, with age-dependent signs and symptoms, in many cases it is difficult to diagnose the disease based on clinical features only.

NGS allows the identification of novel mutations in five genes in the same sequencing run, allowing unambiguous recognition of disorders. It is, therefore, particularly useful to identify genetic pathogenic variant in patients with few symptoms, allowing an appropriate genetic counseling and a personalized follow-up.

## Figures and Tables

**Figure 1 genes-11-00671-f001:**
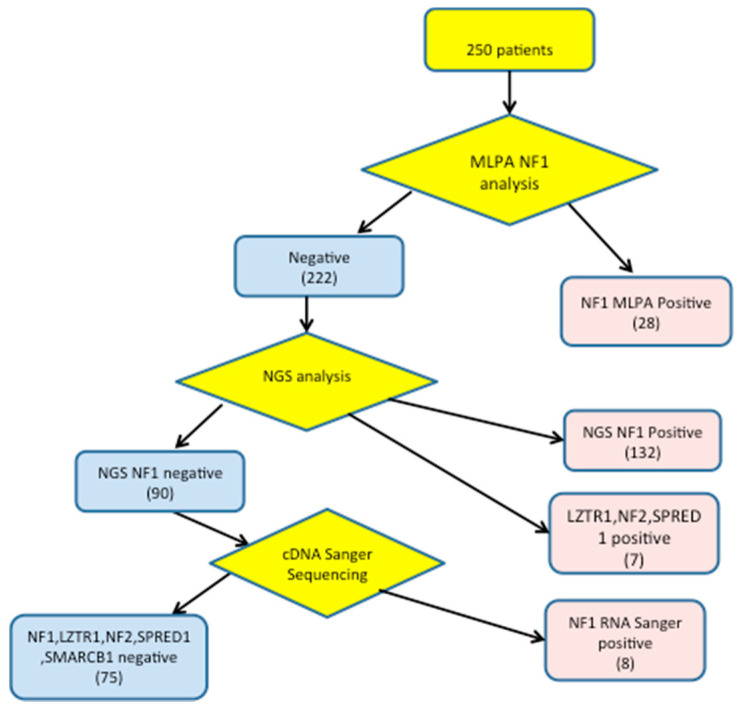
Molecular screening flow chart for patients with suspected NF1.

**Figure 2 genes-11-00671-f002:**
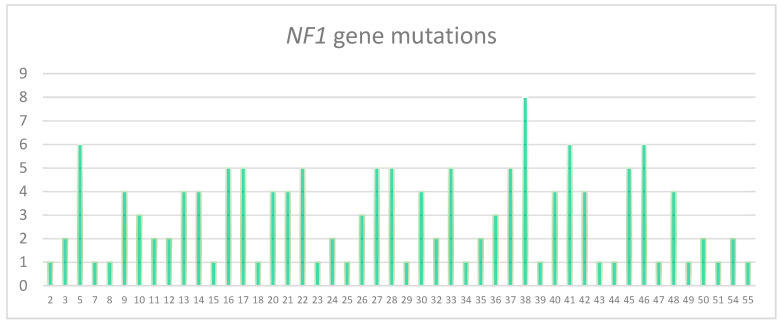
Distributions of the *NF1* single small pathogenic variants identified for each exon.

**Table 1 genes-11-00671-t001:** New *NF1* and *LZTR1* pathogenic variants observed.

Variant On Genome/DBID.	DNA Change	RNA Change	Protein Change
NF1_002811	c.980delT	r.980delu	p.Leu327Argfs*49
NF1_002809	c.269T>G	r.269u>g	p.Leu90Arg
NF1_000806	c.6755A>G	r.6642_6756del115	p.Phe2215fs
NF1_002823	c.6770dupG	r.6770dupg	p.Cys2257Trpfs*6
NF1_002817	c.3574G>T	r.3574g>u	p.Glu1192*
NF1_002812	c.1170delC	r.1170delc	p.Asn390Lysfs*22
NF1_002816	c.2026_2027insT	r.2026_2027insu	p.Thr676fs
NF1_002810	c.586+2 T>G	r.480_586del107	p.Leu161fs
NF1_002819	c.4653delT	r.4653delu	p.Phe1551Leufs*2
NF1_001963	c.5870T>C	r.5870u>c	p.Leu1957Pro
NF1_002826	c.7907+1 _7907+4 delGTAA	r.7807_7907del101	p.Thr2604*
NF1_002818	c.3703C>T	r.3703c>u	p.Gln1235*
NF1_002815	c.1541_1542delAG	r.1541_1542delag	p.Gln514Argfs*43
NF1_002813	c.1185+2delT	r.1063_1185del123	p.Asn355_Lys395del
NF1_002821	c.5199_5205delTATTAAA	r.5199_5205deluauuaaa	p.Ile1734Leufs*8
NF1_002822	c.6756+11 C>T	r.6642_6756del115	p.Phe2215Hisfs*6
NF1_002822	c.6756+11 C>T	r.6642_6756del115	p.Phe2215Hisfs*6
NF1_002824	c.7345_7346delAA	r.7345_7346delaa	p.Asn2449Cysfs*12
NF1_002814	c.1393-3 delTA	r.1393_1527del135	p.Ser465_Cys509del
NF1_002825	c.7433dupG	r.7433dupg	p.Ser2479fs
NF1_002820	c.4773-2A>C	r.4773_5065del293	p.Phe1592Leufs*7
NF1_002834	c.6326_6329delTTCA	r.6326_6329deluuca	p.Ile2109Thrfs*19
NF1_002832	c.4701_4708delAACGTTAA	r.4701_4708delAaacguuaa	p.Thr1568Tyrfs*30
NF1_002827	c.288+1137C>T	r.288_289ins288+1019_288+1136 ins118	p.Gly96Glu97ins39+fs*10
NF1_002829	c.3197+2T>A	r.3114_3197del84	p.Arg1038_Thr1066del
NF1_002828	c.2810T>A	r.2810u>a	p.Leu937*
NF1_002831	c.4684G>T	r.4684g>u	p.Glu1562*
NF1_002830	c.3314+1G>C	r.3275_3314del40	p.Gly1092Aspfs*7
NF1_002833	c.5425delC	r.5425delc	p.Arg1809Alafs*33
NF1_002301	c.2915T>C	r.2915u>c	p.Leu972Pro
NF1_002841	c.5513_5514delTA	r.5513_5514delua	p.Leu1838Serfs*2
NF1_002836	c.1280delC	r.1280delc	p.Pro427Leufs*46
NF1_002838	c.3564_3565delACinsTGA	r.3564_3565delacinsuga	p.Gln1188Hisfs*7
NF1_002840	c.4719_4720dupAC	r.4719_4720dupac	p.Gln1574fs
NF1_002842	c.5989A>C	r.5989a>c	p.Ser1997Arg
NF1_002837	c.3315-8 T>G	r.3315_3496del182	p.Thr1106Leufs*28
NF1_002844	c.7921dupG	r.7921dupg	p.Asp2641fs
NF1_002301	c.2915T>C	r.2915u>c	p.Leu972Pro
NF1_002839	c.3709-9T>A	r.3708_3709insuucucag	p.Asp1237Phefs*4
NF1_002835	c.1260+2 T>G	r.1260_1261insggaaguccaaaag	p.Ser421Glyfs*12
NF1_002843	c.6085G>T	r.6085_6364del280	p.Val2029Lysfs*7
NF1_002861	c.-383_(60+1_61-1)del	r.(?)	p.(?)
NF1_001695	c.(3708+1_3709-1)_(3974+1_3975-1)dup	r.(?)	p.(?)
NF1_002858	c.4435A>G	r.4368_4435del68	p.Phe1457*
NF1_002862	c.1944_1945delAGinsC	r.1944_1945delinsc	p.Glu648Aspfs*40
NF1_002858	c.1122_1125delTCTA	r.1122_1125delucua	p.Asp374Glufs*2
NF1_002858	c.6762_6783delTGAGAGTTGCTTAAAAGGACCT	r.6762_6783del22	p.Glu2255Thrfs*8
NF1_002860	c.1463_1466dupGCTA	r.1463_1466dupgcua	p.Tyr489*
NF1_002859	c.7151_7161delTTGTTGCAAGA	r.7151_7161del	p.Ile2384Asnfs*13
NF1_002858	c.4435A>G	r.4368_4435del68	p.Phe1457*
NF1_002864	c.6005T>A	r.6005u>a	p.Leu2002*
NF1_002863	c.7000-?_8314+?del	r.7000_8314del1314	p.Ser2334Glufs*59
LZTR1_000102	c.844C>T	r.844c>u	p.Gln282*
LZTR1_000103	c.154delC	r.154delc	p.His52Ilefs*49
LZTR1_000041	c.1394C>T	r.1394c>u	p.Ala465Val
LZTR1_000104	c.161G>A	r.161g>a	p.Trp54*

No hot spot region for pathogenic variants was identified in the *NF1* gene (Figure 2).

**Table 2 genes-11-00671-t002:** *NF1* pathogenic variants not observed by NGS and detected on RNA by Sanger.

Mutation Type	Location	*n*°pts
c.288 + 1138 C>T	Intron 3	1
c.499_502 del TGTT	Exon 5	3
c.1021_1022 del GT	Exon 9	1
c.2033 dupC	Exon 18	1
c.4224_4225 del AA insT	Exon 32	1
c.7907 + 791 C>G	Intron 54	1

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
