# Peer review of "Simultaneous Detection of NF1, SPRED1, LZTR1, and NF2 Gene Mutations by Targeted NGS in an Italian Cohort of Suspected NF1 Patients"

_genes, 2020, doi:10.3390/genes11060671_

Round 1
Reviewer 1 Report
The study described in this manuscript uses recent technology (targeted NGS) to describe the incidence of known mutations NF1, SPRED1, LZTR1 and NF2 in people with cafe au lait macules referred for investigation of suspected NF1. This is a comprehensive record of mutation detection in this group of patients. All gene detection was performed on peripheral blood only. Those with mosaic lesions that are undetectable in peripheral blood are left unaccounted for.
As the manuscript stands, the errors of punctuation and grammar are distracting. This must be corrected.
As the authors infer, there is much to be learned regarding the impact of LZTR1 outside of schwannomatosis. Unfortunately, this manuscript does little to provide new information.
Reviewer 2 Report
Bianchessi et al. report their results from a NF1 cohort screen using an amplicon-based NGS panel with NF1 and four additional genes associated with NF1-like phenotypes. 250 patients are studied and a diagnosis is made in 175(?) patients.
The manuscript is generally well written. While the approach is not novel, the authors present results from a moderately-sized gene panel cohort screen and report 52 novel pathogenic variants. The inclusion of cDNA sequencing is interesting and a strong point of the study.
I did notice that parts of the Introduction and large parts of the Methods section were very closely reproduced from a similar 2015 paper about an earlier NF1 cohort by the same group, which was surprisingly not cited at all in this manuscript (Bianchessi et al., Mol Genet Genomic Med, 2015).
Other comments:
- Please provide additional data on the NGS sequencing. What was the average coverage, what % of the target regions was covered at a sufficient depth (e.g. 20x)?
- The detection rates in the different subgroups are unclear to me. In Fig1 (which has a very bad quality for me) the variant detections are split up by method, in the text they are split up by NIH-criteria-positive or -negative subgroups. In Fig1 I see 28 MLPA + 132 NGS + 8 cDNA Sanger-positive patients (=168), in the text there is mention of 132 NIH-criteria-positive and 28 NIH-criteria-negative patients (=160) with detection of pathogenic NF1 variants. Were the cDNA Sanger-positive patients excluded here? Why?
- Why were the "missed by NGS" variants missed? The authors come close to stating this in lines 254-257. Were the regions not covered in the NGS dataset or did the variant caller miss the variants?
- Please use HGVS nomenclature for variant decription consistently throughout the manuscript (e.g. c.1234A>C). Also I suggest using "pathogenic variant" instead of "mutation".
- lines 187-189: How did MLPA detect nonsense- and frameshift-variants? Were these located in the probe binding sequence? I would also suggest a short note about the "type 1/2/3 microdeletion" nomenclature in NF1, which may not be familiar to all readers.
- line 49-50: copy number variation includes deletions, so probably duplications are meant here?
- I was missing a Discussion of the utility of cDNA Sanger sequencing - apprently it prevented 8 false negative results (and increased detection rate by ~5%) but is not routinely done in most laboratories. I think this result should be discussed in some more detail.
- Please italicize all gene names.
- line 173 "no non-pigmentary criteria" is confusing - I suggest rephrasing this.
- line 262: Suggest clarifying that these 5 patients clinically had a "segmental NF" phenotype.
- line 45-46: Suggest rephrasing as "...high mutation rate that is responsible for 50% of all pathogenic variants being de novo".
Round 2
Reviewer 1 Report
Thank you for an improved manuscript.